# Hitting Times for Continuous-Time Imprecise-Markov Chains

**Thomas Krak**[1]

[1]Department of Mathematics and Computer Science, Eindhoven University of Technology, Eindhoven, The Netherlands

## Abstract

We study the problem of characterizing the expected hitting times for a robust generalization of continuous-time Markov chains. This generalization is based on the theory of imprecise probabilities, and the models with which we work essentially constitute sets of stochastic processes. Their inferences are tight lower- and upper bounds with respect to variation within these sets.

We consider three distinct types of these models, corresponding to different levels of generality and structural independence assumptions on the constituent processes.

Our main results are twofold; first, we demonstrate that the hitting times for all three types are equivalent. Moreover, we show that these inferences are described by a straightforward generalization of a well-known linear system of equations that characterizes expected hitting times for traditional time-homogeneous continuous-time Markov chains.

## 1 INTRODUCTION

We consider the problem of characterizing the *expected hitting times* for continuous-time *imprecise-Markov chains* [Škulj, 2015, Krak et al., 2017, Krak, 2021, Erreygers, 2021]. These are *robust*, set-valued generalizations of (traditional) Markov chains [Norris, 1998], based on the theory of *imprecise probabilities* [Walley, 1991, Augustin et al., 2014]. From a sensitivity-analysis perspective, we may interpret these sets as hedging against model-uncertainties with respect to a model's numerical parameters and/or structural (independence) assumptions.

The inference problem of hitting times essentially deals with the question of how long it will take the underlying system to reach some particular subset of its states. This is a common and important problem in such fields as, e.g., reliability analysis, where it can capture the expected time-to-failure of a system; and epidemiology, to model the expected time-until-extinction of an epidemic. For imprecise-Markov chains, then, we are interested in evaluating these quantities in a manner that is robust against, and conservative with respect to, any variation that is compatible with one's uncertainty about the model specification.

Erreygers [2021] has recently obtained some partial results towards characterizing such inferences, but has not been able to give a complete characterization and has largely studied the finite-time horizon case. The problem of hitting times for *discrete-time* imprecise-Markov chains was previously studied by Krak et al. [2019], Krak [2020]. In this present work, we largely emulate and extend their results to the continuous-time setting.

We will be concerned with three different types of imprecise-Markov chains. These are all sets of stochastic processes that are in a specific sense compatible with a given set of numerical parameters, but the three types differ in the independence properties of their elements. In particular, they correspond to (i) a set of (*time-*)*homogeneous* Markov chains, (ii) a set of (not-necessarily homogeneous) Markov chains, and (iii) a set of general—not-necessarily homogeneous nor Markovian—stochastic processes. It is known (and perhaps not very surprising) that inferences with respect to these three models do not in general agree; see e.g. [Krak, 2021] for a detailed analysis of their differences.

However, our first main result in this work is that the expected hitting time is *the same* for these three different types of models. Besides being of theoretical interest, we want to emphasize the power of this result: it means that even if a practitioner using Markov chains would be uncertain whether the system they are studying is truly homogeneous and/or Markovian, relaxing these assumptions would not influence inferences about the hitting times in this sense. Purely pragmatically, it also means that we can use computational methods tailored to any one of these types of models, to compute these inferences.

*Accepted for the 38th Conference on Uncertainty in Artificial Intelligence* (UAI 2022).

Our second main result is that these hitting times are characterized by a generalization of a well-known system of equations that holds for continuous-time homogeneous Markov chains; see Proposition 2 for this linear system.

The remainder of this paper is structured as follows. In Section 2 we introduce the basic required concepts that we will use throughout, formalizing the notion of stochastic processes and defining the inference problem of interest. In Section 3, we define the various types of imprecise-Markov chains that we use throughout this work. We spend some effort in Section 4 to study the transition dynamics of these models, from a perspective that is particularly relevant for the inference problem of hitting times. In Section 5 we explain and sketch the proofs of our main results, and we give a summary in Section 6.

Because we have quite a lot of conceptual material to cover before we can explain our main results, we are not able to fit any real proofs in the main body of this work. Instead, these—together with a number of technical lemmas—have largely been relegated to the supplementary material.

## 2 PRELIMINARIES

Throughout, we consider a fixed, finite *state space* $\mathcal{X}$ with at least two elements. This set contains all possible values for some abstract underlying process. An element of $\mathcal{X}$ is called a *state*, and is usually generically denoted as $x \in \mathcal{X}$.

We use $\mathbb{R}, \mathbb{R}_{\geq 0}$, and $\mathbb{R}_{>0}$ to denote the reals, the non-negative reals, and the positive reals, respectively. $\mathbb{N}$ denotes the natural numbers *without* zero, and we let $\mathbb{N}_0 := \mathbb{N} \cup \{0\}$.

For any $\mathcal{Y} \subseteq \mathcal{X}$, we use $\mathbb{R}^{\mathcal{Y}}$ to denote the vector space of real-valued functions on $\mathcal{Y}$; in particular, $\mathbb{R}^{\mathcal{X}}$ denotes the space of all real functions on $\mathcal{X}$. We use $\|\cdot\|$ to denote the supremum norm on any such space; for any $f \in \mathbb{R}^{\mathcal{Y}}$ we let $\|f\| := \max\{|f(x)| : x \in \mathcal{Y}\}$. Throughout, we make extensive use of *indicator functions*, which are defined for all $A \subseteq \mathcal{Y}$ as $\mathbb{I}_A(x) := 1$ if $x \in A$ and $\mathbb{I}_A(x) := 0$, otherwise. We use the shorthand $\mathbb{I}_y := \mathbb{I}_{\{y\}}$. Let $\mathbf{1}$ denote the function that is identically equal to 1; its dimensionality is to be understood from context.

A map $M : \mathbb{R}^{\mathcal{Y}} \to \mathbb{R}^{\mathcal{Y}}$ is also called an *operator*, and we denote its evaluation in $f \in \mathbb{R}^{\mathcal{Y}}$ as $Mf$. If it holds for all $\lambda \in \mathbb{R}_{\geq 0}$ that $M(\lambda f) = \lambda M f$ then $M$ is called *non-negatively homogeneous*. For any non-negatively homogeneous operator on $\mathbb{R}^{\mathcal{Y}}$, we define the induced operator norm $\|M\| := \sup\{\|Mf\| : f \in \mathbb{R}^{\mathcal{Y}}, \|f\| = 1\}$. We reserve the symbol $I$ to denote the identity operator on any space; the domain is to be understood from context.

Note that any *linear* operator is also non-negatively homogeneous. Moreover, if $M$ is linear it can be represented as an $|\mathcal{Y}| \times |\mathcal{Y}|$ matrix by arbitrarily fixing an ordering on $\mathcal{Y}$. However, without fixing such an ordering,

we simply use $M(x, y) := M\mathbb{I}_y(x)$ to denote the entry in the $x$-row and $y$-column of such a matrix, for any $x, y \in \mathcal{Y}$. For any $f \in \mathbb{R}^{\mathcal{Y}}$ and $x \in \mathcal{Y}$ we then have $Mf(x) = \sum_{y \in \mathcal{Y}} M(x, y) f(y)$, so that $Mf$ simply represents the usual matrix-vector product of $M$ with the (column) vector $f$. In the sequel, we interchangeably refer to linear operators also as matrices. We note the well-known equality $\|M\| = \max_{x \in \mathcal{Y}} \sum_{y \in \mathcal{Y}} |M(x, y)|$ for the induced matrix norm.

### 2.1 PROCESSES & MARKOV CHAINS

We now turn to stochastic processes, which are fundamentally the subject of this work. The typical (measure-theoretic) way to define a stochastic process is simply as a family $(X_i)_{i \in \mathcal{I}}$ of random variables with index set $\mathcal{I}$. This index set represents the time domain of the stochastic process. The random variables are understood to be taken with respect to some underlying probability space $(\Omega_{\mathcal{I}}, \mathcal{F}_{\mathcal{I}}, P)$, where $\Omega_{\mathcal{I}}$ is a set of *sample paths*, which are functions from $\mathcal{I}$ to $\mathcal{X}$ representing possible realizations of the evolution of the underlying process through $\mathcal{X}$. The random variables $X_i, i \in \mathcal{I}$ are canonically the maps $X_i : \omega \mapsto \omega(i)$ on $\Omega_{\mathcal{I}}$.

However, for our purposes it will be more convenient to instead refer to the *probability measure* $P$ as the stochastic process. Different processes $P$ may then be taken over the same measurable space $(\Omega_{\mathcal{I}}, \mathcal{F}_{\mathcal{I}})$, using the same canonical variables $(X_i)_{i \in \mathcal{I}}$ for all these processes.

In this work we will use both *discrete-* and *continuous-time* stochastic processes, which corresponds to choosing $\mathcal{I} = \mathbb{N}_0$ or $\mathcal{I} = \mathbb{R}_{\geq 0}$, respectively. In both cases we take $\mathcal{F}_{\mathcal{I}}$ to be the $\sigma$-algebra generated by the cylinder sets; this ensures that all functions that we consider are measurable.

In the discrete-time case, we let $\Omega_{\mathbb{N}_0}$ be the set of *all* functions from $\mathbb{N}_0$ to $\mathcal{X}$. A discrete-time stochastic process $P$ is then simply a probability measure on $(\Omega_{\mathbb{N}_0}, \mathcal{F}_{\mathbb{N}_0})$. Moreover, $P$ is said to be a *Markov chain* if it satisfies the (discrete-time) *Markov property*, meaning that

$$P(X_{n+1} = x_{n+1} \mid X_0 = x_0, \ldots, X_n = x_n)$$
$$= P(X_{n+1} = x_{n+1} \mid X_n = x_n),$$

for all $x_0, \ldots, x_{n+1} \in \mathcal{X}$ and $n \in \mathbb{N}_0$. If, additionally, it holds for all $x, y \in \mathcal{X}$ and $n \in \mathbb{N}_0$ that

$$P(X_{n+1} = y \mid X_n = x) = P(X_1 = y \mid X_0 = x),$$

then $P$ is said to be a (*time-*)*homogeneous* Markov chain. We use $\mathbb{P}_{\mathbb{N}_0}, \mathbb{P}_{\mathbb{N}_0}^{\mathrm{M}}$, and $\mathbb{P}_{\mathbb{N}_0}^{\mathrm{HM}}$ to denote, respectively, the set of *all* discrete-time stochastic processes; the set of all discrete-time Markov chains; and the set of all discrete-time homogeneous Markov chains.

In the continuous-time case, we let $\Omega_{\mathbb{R}_{\geq 0}}$ be the set of all *cadlag* functions from $\mathbb{R}_{\geq 0}$ to $\mathcal{X}$. A continuous-time stochastic process $P$ is a probability measure on

$(\Omega_{\mathbb{R}_{\geq 0}}, \mathcal{F}_{\mathbb{R}_{\geq 0}})$. The process $P$ is said to be a Markov chain if it satisfies the (continuous-time) Markov property,

$$P(X_{t_{n+1}} = x_{t_{n+1}} \mid X_{t_0} = x_{t_0}, \ldots, X_{t_n} = x_{t_n})$$
$$= P(X_{t_{n+1}} = x_{t_{n+1}} \mid X_{t_n} = x_{t_n})$$

for all $x_{t_0}, \ldots, x_{t_{n+1}} \in \mathcal{X}$, $t_0 < \cdots < t_n \leq t_{n+1} \in \mathbb{R}_{\geq 0}$, and all $n \in \mathbb{N}_0$. If, additionally, it holds that

$$P(X_s = y \mid X_t = x) = P(X_{s-t} = y \mid X_0 = x)$$

for all $x, y \in \mathcal{X}$ and all $t, s \in \mathbb{R}_{\geq 0}$ with $t \leq s$, then $P$ is said to be a (time-)homogeneous Markov chain. We use $\mathbb{P}_{\mathbb{R}_{\geq 0}}, \mathbb{P}^{\mathrm{M}}_{\mathbb{R}_{\geq 0}}$, and $\mathbb{P}^{\mathrm{HM}}_{\mathbb{R}_{\geq 0}}$ to denote, respectively, the set of *all* continuous-time stochastic processes; the set of all continuous-time Markov chains; and the set of all continuous-time homogeneous Markov chains.

We refer to [Norris, 1998] for an excellent further introduction to discrete-time and continuous-time Markov chains.

## 2.2 TRANSITION DYNAMICS

Throughout this work, we make extensive use of operator-theoretic representations of the behavior of stochastic processes, and Markov chains in particular. The first reason for this is that such operators serve as a way to parameterize Markov chains. Moreover, they are also useful as a *computational* tool, since they can often be used to express inferences of interest; see, e.g., Propositions 1 and 2 further on. We introduce the basic concepts below, and refer to e.g. [Norris, 1998] for details.

A *transition matrix* $T$ is a linear operator on $\mathbb{R}^{\mathcal{X}}$ such that, for all $x \in \mathcal{X}$, it holds that $T(x, y) \geq 0$ for all $y \in \mathcal{X}$, and $\sum_{y \in \mathcal{X}} T(x, y) = 1$. There is an important and well-known connection between Markov chains and transition matrices; for any discrete-time homogeneous Markov chain $P$, we can define the *corresponding transition matrix* ${}^P T$ as

$$ {}^P T(x, y) := P(X_1 = y \mid X_0 = x) \quad \text{for all } x, y \in \mathcal{X}.$$

Since $P$ is a probability measure, we clearly have that ${}^P T$ is a transition matrix. Conversely, a given transition matrix $T$ uniquely determines a discrete-time homogeneous Markov chain $P$ with ${}^P T = T$, up to the specification of the initial distribution $P(X_0)$. For this reason, transition matrices are often taken as a crucial parameter to specify (discrete-time, homogeneous) Markov chains.

Analogously, for a (non-homogeneous) discrete-time Markov chain $P$, we might define a family $({}^P T_n)_{n \in \mathbb{N}_0}$ of *time-dependent* corresponding transition matrices, with

$$ {}^P T_n(x, y) := P(X_{n+1} = y \mid X_n = x),$$

for all $x, y \in \mathcal{X}$ and $n \in \mathbb{N}_0$. Conversely, any family $(T_n)_{n \in \mathbb{N}_0}$ of transition matrices uniquely determines

a discrete-time Markov chain $P$ with ${}^P T_n = T_n$ for all $n \in \mathbb{N}_0$, again up to the specification of $P(X_0)$.

In the continuous-time setting, transition matrices are also of great importance. However, it will be instructive to first introduce rate matrices. A *rate matrix* $Q$ is a linear operator on $\mathbb{R}^{\mathcal{X}}$ such that, for all $x \in \mathcal{X}$, it holds that $Q(x, y) \geq 0$ for all $y \in \mathcal{X}$ with $x \neq y$, and $\sum_{y \in \mathcal{X}} Q(x, y) = 0$.

For any rate matrix $Q$ and any $t \in \mathbb{R}_{\geq 0}$, the *matrix exponential* $e^{Qt}$ of $Qt$ can be defined as [Van Loan, 2006]

$$ e^{Qt} := \lim_{n \to +\infty} \left( I + {}^t\!/nQ \right)^n.$$

An alternative characterization is as the (unique) solution to the matrix ordinary differential equation [Van Loan, 2006]

$$ \frac{\mathrm{d}}{\mathrm{d}s} e^{Qs} = Q e^{Qs} = e^{Qs} Q, \quad \text{with } e^{Q0} = I. \quad (1)$$

For any $t, s \in \mathbb{R}_{\geq 0}$ it holds that $e^{Q(t+s)} = e^{Qt} e^{Qs}$, and we immediately have $e^{Q0} = I$. The family $(e^{Qt})_{t \in \mathbb{R}_{\geq 0}}$ is therefore called the *semigroup* generated by $Q$, and $Q$ is called the *generator* of this semigroup. Moreover, for any rate matrix $Q$ and any $t \in \mathbb{R}_{\geq 0}$, $e^{Qt}$ is a transition matrix [Norris, 1998, Thm 2.1.2].

Now let us consider a continuous-time homogeneous Markov chain $P$, and define the corresponding transition matrix[1] ${}^P T_t$ for all $t \in \mathbb{R}_{\geq 0}$ and $x, y \in \mathcal{X}$ as

$$ {}^P T_t(x, y) := P(X_t = y \mid X_0 = x). \quad (2)$$

It turns out that there is then a unique rate matrix ${}^P Q$ associated with $P$ such that ${}^P T_t = e^{{}^P Q t}$ for all $t \in \mathbb{R}_{\geq 0}$. By combining Equations (1) and (2), we can identify ${}^P Q$ as

$$ {}^P Q = \left( \frac{\mathrm{d}}{\mathrm{d}t} {}^P T_t \right) \bigg|_{t=0}.$$

As before, in the other direction we have that any fixed rate matrix $Q$ uniquely determines a continuous-time homogeneous Markov chain $P$ with ${}^P Q = Q$, up to the specification of $P(X_0)$. For this reason, rate matrices are often used to specify (continuous-time, homogeneous) Markov chains.

Let us finally consider a (not-necessarily homogeneous) continuous-time Markov chain $P$. For any $t, s \in \mathbb{R}_{\geq 0}$ with $t \leq s$, we can then define a transition matrix ${}^P T_t^s$ with, for all $x, y \in \mathcal{X}$, ${}^P T_t^s(x, y) := P(X_s = y \mid X_t = x)$. Under appropriate assumptions of differentiability, this induces a family $({}^P Q_t)_{t \in \mathbb{R}_{\geq 0}}$ of rate matrices ${}^P Q_t$, as

$$ {}^P Q_t = \left( \frac{\mathrm{d}}{\mathrm{d}s} {}^P T_t^s \right) \bigg|_{s=t}. \quad (3)$$

In the converse direction we might try to reconstruct the transition matrices of $P$ by solving the matrix ordinary

---

[1] Note that in continuous-time, we always have to measure the transition-time interval $[0, t]$ to specify these matrices.

differential equation(s)

$$\frac{\mathrm{d}}{\mathrm{d}\,s}\,{}^P T_t^s = {}^P T_t^s\, {}^P Q_s, \quad \text{with } {}^P T_t^t = I. \qquad (4)$$

By comparing with Equation (1), we see that in the special case where ${}^P Q_s$ does not depend on $s$—that is, where $P$ is homogeneous with ${}^P Q_s = {}^P Q$, say—we indeed obtain ${}^P T_t^s = e^{{}^P Q(s-t)}$. However, in general the *non-autonomous* system (4) does not have such a closed-form solution, and we cannot move beyond this implicit characterization.

## 2.3   HITTING TIMES

We now have all the pieces to introduce the inference problem that is the subject of this work, *viz.* the *expected hitting times* of some non-empty set of states $A \subset \mathcal{X}$ with respect to a particular stochastic process. We take this set $A$ to be fixed for the remainder of this work.

In the discrete-time case, we consider the (extended real-valued)[2] function $\tau_{\mathbb{N}_0} : \Omega_{\mathbb{N}_0} \to \mathbb{R}_{\geq 0} \cup \{+\infty\}$ given by

$$\tau_{\mathbb{N}_0}(\omega) := \inf\big\{n \in \mathbb{N}_0 \,:\, \omega(n) \in A\big\} \quad \text{for all } \omega \in \Omega_{\mathbb{N}_0}.$$

This captures the number of steps before a process $P$ "hits" any state in $A$. The expected hitting time for a discrete-time process $P$ starting in $x \in \mathcal{X}$ is then defined as

$$\mathbb{E}_P\big[\tau_{\mathbb{N}_0} \,|\, X_0 = x\big] := \int_{\Omega_{\mathbb{N}_0}} \tau_{\mathbb{N}_0}(\omega)\,\mathrm{d}P(\omega \,|\, X_0 = x)\,.$$

We use $\mathbb{E}_P\big[\tau_{\mathbb{N}_0} \,|\, X_0\big]$ to denote the extended real-valued function on $\mathcal{X}$ given by $x \mapsto \mathbb{E}_P\big[\tau_{\mathbb{N}_0} \,|\, X_0 = x\big]$. When dealing with homogeneous Markov chains, this quantity has the following simple characterization:

**Proposition 1.** *[Norris, 1998, Thm 1.3.5] Let $P$ be a discrete-time homogeneous Markov chain with corresponding transition matrix ${}^P T$. Then $h := \mathbb{E}_P\big[\tau_{\mathbb{N}_0} \,|\, X_0\big]$ is the minimal non-negative solution to the linear system*[34]

$$h = \mathbb{I}_{A^c} + \mathbb{I}_{A^c}\,{}^P T h\,.$$

In the continuous-time case, the definition is analogous; we introduce a function $\tau_{\mathbb{R}_{\geq 0}} : \Omega_{\mathbb{R}_{\geq 0}} \to \mathbb{R}_{\geq 0} \cup \{+\infty\}$ as

$$\tau_{\mathbb{R}_{\geq 0}}(\omega) := \inf\big\{t \in \mathbb{R}_{\geq 0} \,:\, \omega(t) \in A\big\} \text{ for all } \omega \in \Omega_{\mathbb{R}_{\geq 0}}.$$

This function measures the time until a process "hits" any state in $A$ on a given sample path. The expected hitting time

for a continuous-time process $P$ starting in $x \in \mathcal{X}$ is

$$\mathbb{E}_P\big[\tau_{\mathbb{R}_{\geq 0}} \,|\, X_0 = x\big] := \int_{\Omega_{\mathbb{R}_{\geq 0}}} \tau_{\mathbb{R}_{\geq 0}}(\omega)\,\mathrm{d}P(\omega \,|\, X_0 = x)\,.$$

We again use $\mathbb{E}_P\big[\tau_{\mathbb{R}_{\geq 0}} \,|\, X_0\big]$ to denote the extended-real valued function on $\mathcal{X}$ given by $x \mapsto \mathbb{E}_P\big[\tau_{\mathbb{R}_{\geq 0}} \,|\, X_0 = x\big]$. Also in this case, the characterization for homogeneous Markov chains is particularly simple:

**Proposition 2.** *[Norris, 1998, Thm 3.3.3] Let $P$ be a continuous-time homogeneous Markov chain with rate matrix ${}^P Q$ such that ${}^P Q(x, x) \neq 0$ for all $x \in A^c$. Then $h := \mathbb{E}_P\big[\tau_{\mathbb{R}_{\geq 0}} \,|\, X_0\big]$ is the minimal non-negative solution to*

$$\mathbb{I}_A h = \mathbb{I}_{A^c} + \mathbb{I}_{A^c}\,{}^P Q h\,. \qquad (5)$$

# 3   IMPRECISE-MARKOV CHAINS

Let us now introduce *imprecise-Markov chains* [Hermans and Škulj, 2014, Škulj, 2015, Krak et al., 2017], which are the stochastic processes that we aim to study in this work. Their characterization is based on the theory of *imprecise probabilities* [Walley, 1991, Augustin et al., 2014].

We here adopt the "sensitivity analysis" interpretation of imprecise probabilities. This means that we represent an imprecise-Markov chain simply as a *set* $\mathcal{P}$ of stochastic processes. Intuitively, the idea is that we collect in $\mathcal{P}$ all (traditional, "precise") stochastic processes that we deem to plausibly capture the dynamics of the underlying system of interest. Inferences with respect to $\mathcal{P}$ are defined using *lower*- and *upper* expectations, given respectively as

$$\underline{\mathbb{E}}_{\mathcal{P}}[\cdot \,|\, \cdot] := \inf_{P \in \mathcal{P}} \mathbb{E}_P[\cdot \,|\, \cdot] \quad \text{and} \quad \overline{\mathbb{E}}_{\mathcal{P}}[\cdot \,|\, \cdot] := \sup_{P \in \mathcal{P}} \mathbb{E}_P[\cdot \,|\, \cdot]\,.$$

So, their inferences represent *robust*—i.e. conservative—and *tight* lower- and upper bounds on inferences with respect to *all* stochastic processes that we deem to be plausible.

## 3.1   SETS OF PROCESSES & TYPES

We already mentioned that an imprecise-Markov chain is essentially simply a set $\mathcal{P}$ of stochastic processes. Let us now consider how to define such sets.

We start by considering the discrete-time case; then, clearly, $\mathcal{P}$ will be a set of discrete-time processes. We will parameterize such a set with some non-empty set $\mathcal{T}$ of transition matrices. Our aim is then to include in $\mathcal{P}$ all processes that are in some sense "compatible" with $\mathcal{T}$.[5] However, at this point we are faced with a choice about which *type* of processes to include in this set, and these different choices lead to *different types of imprecise-Markov chains*.

---

[2]We agree that $0(+\infty) = 0$; $(+\infty) + (+\infty) = +\infty$; and, for any $c \in \mathbb{R}$, $(+\infty) + c = +\infty$ and $c(+\infty) = +\infty$ if $c > 0$.

[3]Throughout, for any $f, g \in \mathbb{R}^{\mathcal{X}}$, the quantity $fg$ is understood as the pointwise product between the functions $f$ and $g$.

[4]Strictly speaking this requires extending the domain of ${}^P T$ to extended-real valued functions, but we will shortly introduce some assumptions that obviate such an exposition.

[5]We will not constrain the initial models $P(X_0)$ of the elements of $\mathcal{P}$, since in any case such a choice would not influence the inferences that we study in this work.

Arguably the conceptually most simple model is $\mathcal{P}_{\mathcal{T}}^{\mathrm{HM}}$, which contains all homogeneous Markov chains $P$ whose corresponding transition matrix is included in $\mathcal{T}$:

$$\mathcal{P}_{\mathcal{T}}^{\mathrm{HM}} := \left\{ P \in \mathbb{P}_{\mathbb{N}_0}^{\mathrm{HM}} \ : \ {}^P T \in \mathcal{T} \right\}.$$

However, we could instead consider $\mathcal{P}_{\mathcal{T}}^{\mathrm{M}}$, which is the set of all (not-necessarily homogeneous) Markov chains whose time-dependent transition matrices are contained in $\mathcal{T}$:

$$\mathcal{P}_{\mathcal{T}}^{\mathrm{M}} := \left\{ P \in \mathbb{P}_{\mathbb{N}_0}^{\mathrm{M}} \ : \ {}^P T_n \in \mathcal{T} \text{ for all } n \in \mathbb{N}_0 \right\}.$$

The last choice that we consider here is the set $\mathcal{P}_{\mathcal{T}}^{\mathrm{I}}$, which essentially contains *all* discrete-time processes whose single-step transition dynamics are described by $\mathcal{T}$. Its characterization is more cumbersome since we have not expressed these general processes in terms of transition matrices, but we can say that it is the set of all $P \in \mathbb{P}_{\mathbb{N}_0}$ such that for all $n \in \mathbb{N}_0$ and all $x_0, \ldots, x_n \in \mathcal{X}$, there is some $T \in \mathcal{T}$ such that for all $y \in \mathcal{X}$ it holds that

$$P(X_{n+1} = y \,|\, X_0 = x_0, \ldots, X_n = x_n) = T(x_n, y).$$

This last type is called an imprecise-Markov chain under *epistemic irrelevance*, whence the superscript 'I'.

Note that the three types $\mathcal{P}_{\mathcal{T}}^{\mathrm{HM}}, \mathcal{P}_{\mathcal{T}}^{\mathrm{M}}$, and $\mathcal{P}_{\mathcal{T}}^{\mathrm{I}}$ capture not only "plausible" variation in terms of parameter uncertainty—expressed through the set $\mathcal{T}$—but also variation in terms of the structural independence conditions that we consider! So, from an applied perspective, if someone is not sure whether the underlying system that they are studying is truly Markovian and/or time-homogeneous, they might choose to use different such sets in their analysis.

In the continuous-time case, we again proceed analogously. First, we fix a non-empty set $\mathcal{Q}$ of rate matrices, which will be the parameter for our models. We then first consider the set $\mathcal{P}_{\mathcal{Q}}^{\mathrm{HM}}$ of all homogeneous Markov chains whose rate matrix is included in $\mathcal{Q}$:

$$\mathcal{P}_{\mathcal{Q}}^{\mathrm{HM}} := \left\{ P \in \mathbb{P}_{\mathbb{R}_{\geq 0}}^{\mathrm{HM}} \ : \ {}^P Q \in \mathcal{Q} \right\}.$$

The other two types are constructed in analogy to the discrete-time case, but unfortunately we don't have the space for a complete exposition of their characterization. Instead we refer the interested reader to [Krak et al., 2017, Krak, 2021] for an in-depth study of these different types and comparisons between them; in what follows we limit ourselves to a largely intuitive specification.

The model $\mathcal{P}_{\mathcal{Q}}^{\mathrm{M}}$ is the set of all continuous-time (not-necessarily homogeneous) Markov chains whose transition dynamics are compatible with $\mathcal{Q}$ at every point in time. This includes in particular all Markov chains $P$ satisfying the appropriate differentiability assumptions to meaningfully say that the time-dependent rate matrices ${}^P Q_t$—as in Equation (3)—are included in $\mathcal{Q}$ for all $t \in \mathbb{R}_{\geq 0}$. However, $\mathcal{P}_{\mathcal{Q}}^{\mathrm{M}}$

also contains other processes that are not (everywhere) differentiable; see e.g. [Krak, 2021, Sec 4.6 and 5.2] for the technical details.

The most involved model to explain is again $\mathcal{P}_{\mathcal{Q}}^{\mathrm{I}}$, which includes *all* continuous-time processes whose time- and history-dependent transition dynamics can be described using elements of $\mathcal{Q}$. It includes, but is not limited to, appropriately differentiable processes $P$ such that for all $n \in \mathbb{N}_0$, all $t_0 < \cdots < t_n \in \mathbb{R}_{\geq 0}$, and all $x_{t_0}, \ldots, x_{t_n} \in \mathcal{X}$, there is some $Q \in \mathcal{Q}$ such that for all $y \in \mathcal{X}$ it holds that

$$\left( \frac{\mathrm{d}}{\mathrm{d}s} P(X_s = y \,|\, X_{t_0} = x_{t_0}, \ldots, X_{t_n} = x_{t_n}) \right) \Bigg|_{s=t_n}$$
$$= Q(x_{t_n}, y)$$

We again refer to [Krak, 2021, Sec 4.6 and 5.2] for the technical details involving the additional elements of $\mathcal{P}_{\mathcal{Q}}^{\mathrm{I}}$ that are not appropriately differentiable. Importantly, we note the nested structure [Krak, 2021, Prop 5.9]

$$\mathcal{P}_{\mathcal{Q}}^{\mathrm{HM}} \subseteq \mathcal{P}_{\mathcal{Q}}^{\mathrm{M}} \subseteq \mathcal{P}_{\mathcal{Q}}^{\mathrm{I}},$$

where the inclusions are strict provided $\mathcal{Q}$ isn't trivial.

For notational convenience, we will use identical sub- and superscripts to denote the corresponding lower- and upper expectations for any of these imprecise-Markov chains; e.g., we let $\mathbb{E}_{\mathcal{T}}^{\mathrm{HM}}[\cdot \,|\, \cdot] := \mathbb{E}_{\mathcal{P}_{\mathcal{T}}^{\mathrm{HM}}}[\cdot \,|\, \cdot]$.

## 3.2 IMPRECISE TRANSITION DYNAMICS

Let us now introduce some machinery to describe the dynamics of imprecise-Markov chains. In particular, we here move from the set-valued parameters $\mathcal{T}$ and $\mathcal{Q}$ used in Section 3.1, to their dual representations; these are operators that can serve as computational tools.

In Section 3.1, we described discrete-time imprecise-Markov chains using non-empty sets $\mathcal{T}$ of transition matrices. With any such set, we can associate the corresponding *lower-* and *upper transition operators* $\underline{T}$ and $\overline{T}$ on $\mathbb{R}^{\mathcal{X}}$, defined respectively as

$$\underline{T}f := \inf_{T \in \mathcal{T}} Tf \quad \text{and} \quad \overline{T}f := \sup_{T \in \mathcal{T}} Tf \quad \text{for all } f \in \mathbb{R}^{\mathcal{X}}.$$

More generally, any operator $\underline{T}$ (resp. $\overline{T}$) on $\mathbb{R}^{\mathcal{X}}$ is a *lower* (resp. *upper*) *transition operator* if for all $f, g \in \mathbb{R}^{\mathcal{X}}$, all $\lambda \in \mathbb{R}_{\geq 0}$, and all $x \in \mathcal{X}$, it holds that [De Bock, 2017]

1. $\min_{y \in \mathcal{X}} f(y) \leq \underline{T}f(x)$ and $\overline{T}f(x) \leq \max_{y \in \mathcal{X}} f(y)$
2. $\underline{T}f + \underline{T}g \leq \underline{T}(f + g)$ and $\overline{T}(f + g) \leq \overline{T}f + \overline{T}g$
3. $\underline{T}(\lambda f) = \lambda \underline{T}f$ and $\overline{T}(\lambda f) = \lambda \overline{T}f$.

It should be noted that lower- and upper transition operators are conjugate, in that any $\underline{T}$ induces a corresponding upper

transition operator $\overline{T}(\cdot) = -\underline{T}(-\cdot)$, and *vice versa*. Moreover, any transition matrix $T$ is also a lower—and, by its linearity, upper—transition operator.

It is easily verified that the lower- and upper transition operators corresponding to a given non-empty set $\mathcal{T}$ are, indeed, lower- and upper transition operators. Conversely, with a given lower transition operator $\underline{T}$, we can associate the set of transition matrices that *dominate* it, in the sense that

$$\mathcal{T}_{\underline{T}} := \left\{ T \,:\, T \text{ a trans. mat., } Tf \geq \underline{T}f \text{ for all } f \in \mathbb{R}^{\mathcal{X}} \right\}.$$

This set satisfies the following important properties:

**Proposition 3.** *[Krak, 2021, Sec 3.4] Let $\underline{T}$ be a lower transition operator with conjugate upper transition operator $\overline{T}(\cdot) = -\underline{T}(-\cdot)$ and dominating set of transition matrices $\mathcal{T}_{\underline{T}}$. Then $\mathcal{T}_{\underline{T}}$ is a non-empty, closed, and convex set of transition matrices that has separately specified rows,[6] and for all $f \in \mathbb{R}^{\mathcal{X}}$ it holds that $\underline{T}f = \inf_{T \in \mathcal{T}_{\underline{T}}} Tf$ and $\overline{T}f = \sup_{T \in \mathcal{T}_{\underline{T}}} Tf$. Moreover, for all $f \in \mathbb{R}^{\mathcal{X}}$ there is some $T \in \mathcal{T}_{\underline{T}}$ such that $Tf = \underline{T}f$, and there is some—possibly different—$T \in \mathcal{T}_{\underline{T}}$ such that $Tf = \overline{T}f$.*

Notably, there is a one-to-one relation between non-empty sets of transition matrices that are closed and convex and have separately specified rows, and lower (or upper) transition operators: if $\underline{T}$ is the lower transition operator for the set $\mathcal{T}$, and if $\mathcal{T}$ satisfies these properties, then $\mathcal{T} = \mathcal{T}_{\underline{T}}$ [Krak, 2021, Cor 3.38]. Hence these objects may serve as dual representations for each other.

One reason that this is important is the use of $\underline{T}$ as a computational tool; under the conditions of this duality it holds that for any function $f \in \mathbb{R}^{\mathcal{X}}$ and any $n \in \mathbb{N}_0$, we can write [Hermans and Škulj, 2014]

$$\underline{\mathbb{E}}_{\mathcal{T}}^{\mathrm{I}}[f(X_n)|X_0 = x] = \underline{\mathbb{E}}_{\mathcal{T}}^{\mathrm{M}}[f(X_n)|X_0 = x] = \underline{T}^n f(x),$$

where $\underline{T}$ is the lower transition operator for $\mathcal{T}$. This reduces the problem of computing such inferences for the imprecise-Markov chains $\mathcal{P}_{\mathcal{T}}^{\mathrm{M}}$ and $\mathcal{P}_{\mathcal{T}}^{\mathrm{I}}$ to solving $n$ independent *linear optimization problems* over $\mathcal{T}$; first compute $f_1 := \underline{T}f$, then compute $f_2 := \underline{T}f_1 = \underline{T}^2 f$, and so forth. Note that this method in general only yields a conservative bound on the corresponding inference for $\mathcal{P}_{\mathcal{T}}^{\mathrm{HM}}$, as the minimizers $T_k$ that obtain $T_k f_{k-1} = \underline{T}f_{k-1}$ may be different at each step.

We next consider the dynamics in the continuous-time setting. We proceed analogously to the above: we first consider a non-empty and bounded[7] set $\mathcal{Q}$ of rate matrices. With this set, we then associate the corresponding *lower-* and *upper rate operators* $\underline{Q}$ and $\overline{Q}$ on $\mathbb{R}^{\mathcal{X}}$, defined as

$$\underline{Q}f := \inf_{Q \in \mathcal{Q}} Qf \quad \text{and} \quad \overline{Q}f := \sup_{Q \in \mathcal{Q}} Qf \quad \text{for all } f \in \mathbb{R}^{\mathcal{X}}.$$

---

[6]A set $\mathcal{M}$ of matrices is said to have *separately specified rows* if, intuitively, it is closed under the row-wise recombination of its elements; see e.g. [Hermans and Škulj, 2014] for details.

[7]In the induced operator norm.

More generally, any operator $\underline{Q}$ (resp. $\overline{Q}$) on $\mathbb{R}^{\mathcal{X}}$ is a *lower* (resp. *upper*) *rate operator* if for all $f, g \in \mathbb{R}^{\mathcal{X}}$, all $\lambda \in \mathbb{R}_{\geq 0}$ and $\mu \in \mathbb{R}$, and all $x, y \in \mathcal{X}$ with $y \neq x$, it holds that [De Bock, 2017]

1. $\underline{Q}(\mu\mathbf{1})(x) = 0$ and $\overline{Q}(\mu\mathbf{1})(x) = 0$
2. $\underline{Q}\mathbb{I}_y(x) \geq 0$ and $\overline{Q}\mathbb{I}_y(x) \geq 0$
3. $\underline{Q}f + \underline{Q}g \leq \underline{Q}(f + g)$ and $\overline{Q}(f + g) \leq \overline{Q}f + \overline{Q}g$
4. $\underline{Q}(\lambda f) = \lambda\underline{Q}f$ and $\overline{Q}(\lambda f) = \lambda\overline{Q}f$

As before, such objects are conjugate, in that if $\underline{Q}$ is a lower rate operator, then $\overline{Q}(\cdot) = -\underline{Q}(-\cdot)$ is an upper rate operator. Moreover, any rate matrix $Q$ is also a lower (and upper) rate operator. There is again a duality between lower (or upper) rate operators, and sets of rate matrices. For fixed $\underline{Q}$ and with the dominating set of rate matrices $\mathcal{Q}_{\underline{Q}}$ defined as

$$\mathcal{Q}_{\underline{Q}} := \left\{ Q \,:\, Q \text{ a rate mat., } Qf \geq \underline{Q}f \text{ for all } f \in \mathbb{R}^{\mathcal{X}} \right\},$$

we have the following result:

**Proposition 4.** *[Krak, 2021, Sec 6.2] Let $\underline{Q}$ be a lower rate operator with conjugate upper rate operator $\overline{Q}(\cdot) = -\underline{Q}(-\cdot)$ and dominating set of rate matrices $\mathcal{Q}_{\underline{Q}}$. Then $\mathcal{Q}_{\underline{Q}}$ is a non-empty, compact, and convex set of rate matrices that has separately specified rows, and for all $f \in \mathbb{R}^{\mathcal{X}}$ it holds that $\underline{Q}f = \inf_{Q \in \mathcal{Q}_{\underline{Q}}} Qf$ and $\overline{Q}f = \sup_{Q \in \mathcal{Q}_{\underline{Q}}} Qf$. Moreover, for all $f \in \mathbb{R}^{\mathcal{X}}$ there is some $Q \in \mathcal{Q}_{\underline{Q}}$ such that $Qf = \underline{Q}f$, and there is some—possibly different—$Q \in \mathcal{Q}_{\underline{Q}}$ such that $Qf = \overline{Q}f$.*

Now fix any lower rate operator $\underline{Q}$ and any $t \in \mathbb{R}_{\geq 0}$, and let

$$e^{\underline{Q}t} := \lim_{n \to +\infty} \left( I + {}^{t}\!/{}_{n}\underline{Q} \right)^n. \tag{6}$$

The operator $e^{\underline{Q}t}$ is then a lower transition operator [De Bock, 2017], and the family $(e^{\underline{Q}t})_{t \in \mathbb{R}_{\geq 0}}$ is a semigroup of lower transition operators; it satisfies $e^{\underline{Q}(t+s)} = e^{\underline{Q}t}e^{\underline{Q}s}$ for all $t, s \in \mathbb{R}_{\geq 0}$, and $e^{\underline{Q}0} = I$. The analogous construction with an upper rate operator $\overline{Q}$ instead generates a semigroup $(e^{\overline{Q}t})_{t \in \mathbb{R}_{\geq 0}}$ of upper transition operators. When $\underline{Q}$ and $\overline{Q}$ are taken with respect to the same set $\mathcal{Q}$, these semigroups satisfy, for all $t \in \mathbb{R}_{\geq 0}$, $f \in \mathbb{R}^{\mathcal{X}}$, and $Q \in \mathcal{Q}$,

$$e^{\underline{Q}t}f \leq e^{Qt}f \leq e^{\overline{Q}t}f. \tag{7}$$

Here the importance again derives from the use as a computational tool; under the conditions of duality between $\mathcal{Q}$ and $\underline{Q}$, we have for any $f \in \mathbb{R}^{\mathcal{X}}$ and any $t \in \mathbb{R}_{\geq 0}$ that [Škulj, 2015, Krak et al., 2017]

$$\underline{\mathbb{E}}_{\mathcal{Q}}^{\mathrm{I}}[f(X_t)|X_0 = x] = \underline{\mathbb{E}}_{\mathcal{Q}}^{\mathrm{M}}[f(X_t)|X_0 = x] = e^{\underline{Q}t}f(x).$$

Hence such inferences can be numerically computed by approximating (6) with a finite choice of $n$, and then solving $n$ independent linear optimization problems over $\mathcal{Q}$. Error bounds for this scheme are available in the literature [Škulj, 2015, Krak et al., 2017, Erreygers, 2021].

## 3.3 CLASS STRUCTURE

Let us now fix a set $\mathcal{Q}$ of rate matrices that we will use in the remainder of this work. Throughout, let $\underline{Q}$ and $\overline{Q}$ denote the lower- and upper rate operators associated with $\mathcal{Q}$. We impose several standard regularity conditions on this set: we assume that $\mathcal{Q}$ is non-empty, compact, convex, and that it has separately specified rows. These are common assumptions that are imposed to ensure the duality between $\mathcal{Q}$ and $\underline{Q}$, which in turn guarantees that inferences with the induced imprecise-Markov chains remain well-behaved, as well as analytically (and, often, computationally) tractable.

We now have all the pieces to start studying the inference problem that is the subject of this work: the *lower-* and *upper expected hitting times* of the set $A \subset \mathcal{X}$ for *continuous-time imprecise-Markov chains described by $\mathcal{Q}$*.

Before we begin, let us impose two additional conditions on the dynamics of the system.

**Assumption 1.** We assume that all states in $A$ are *absorbing*, which is equivalent to requiring that $Q(x, x) = 0$ for all $Q \in \mathcal{Q}$ and all $x \in A$.

Note that this does not influence the inferences in which we are interested, since those only deal with behavior at times *before* states in $A$ are reached. However, imposing this explicitly substantially simplifies the analysis.

Next, we assume that the set $A$ is *lower reachable* from any state $x \in A^c$ [De Bock, 2017]. This means that we can construct a sequence $x_1, \ldots, x_{n+1} \in \mathcal{X}$ starting in any $x_1 \in A^c$ and ending in some $x_{n+1} \in A$ such that, for all $k = 1, \ldots, n$, it holds that $\underline{Q}\mathbb{I}_{x_{k+1}}(x_k) > 0$. This is equivalent [De Bock, 2017] to

**Assumption 2.** We assume $e^{\underline{Q}t}\mathbb{I}_A(x) > 0$ for all $t \in \mathbb{R}_{>0}$ and all $x \in A^c$.

Essentially, this means that for all elements of our imprecise-probabilistic models the probability of eventually hitting $A$ is bounded away from zero. This ensures that the expected hitting times remain bounded for all $P \in \mathcal{P}_{\mathcal{Q}}^{\mathrm{I}}$, so that we can ignore any extended real-valued analysis. It also implies that for all $Q \in \mathcal{Q}$ we have that $Q(x, x) \neq 0$ for all $x \in A^c$, which is relevant to meet the precondition of Proposition 2. As a practical point, De Bock [2017] gives an algorithm to check whether a given set $\mathcal{Q}$ satisfies this condition.

On a technical level, Assumption 2 is the crucial one for our results, and—unlike with Assumption 1—it cannot really be ignored in practice. However, based on earlier work by Krak et al. [2019] in the discrete-time setting, we hope in the future to strengthen our results to hold without this assumption.

## 4 SUBSPACE DYNAMICS

In the context of hitting times, the interesting behavior of a process actually occurs *before* it has reached a target state in $A$. Hence it will be useful to introduce some machinery to study the transition dynamics as it relates to the states $A^c$.

To introduce the notation in a general way, choose any non-empty $\mathcal{Y} \subset \mathcal{X}$. Then for any $f \in \mathbb{R}^{\mathcal{X}}$, let $f|_{\mathcal{Y}} \in \mathbb{R}^{\mathcal{Y}}$ denote the restriction of $f$ to $\mathcal{Y}$. Conversely, for any $f \in \mathbb{R}^{\mathcal{Y}}$, let $f\uparrow_{\mathcal{X}} \in \mathbb{R}^{\mathcal{X}}$ denote the unique extension of $f$ to $\mathcal{X}$ that satisfies $f(x) = 0$ for all $x \in \mathcal{X} \setminus \mathcal{Y}$. Moreover, for any operator $M$ on $\mathbb{R}^{\mathcal{X}}$, we define the operator $M|_{\mathcal{Y}}$ on $\mathbb{R}^{\mathcal{Y}}$ as

$$M|_{\mathcal{Y}}f := \big(M(f\uparrow_{\mathcal{X}})\big)|_{\mathcal{Y}} \qquad \text{for all } f \in \mathbb{R}^{\mathcal{Y}}.$$

This somewhat verbose notation is perhaps most easily understood when $M$ is a linear operator, i.e. a matrix. In that case, $M|_{\mathcal{Y}}$ is simply the $|\mathcal{Y}| \times |\mathcal{Y}|$ sub-matrix of $M$ on the coordinates in $\mathcal{Y}$. The definition above allows us to extend this notion also to non-linear operators, and to lower- and upper transition and rate operators, specifically.

Now for any rate matrix $Q \in \mathcal{Q}$, we call $G := Q|_{A^c}$ its corresponding *subgenerator*. For any $t \in \mathbb{R}_{\geq 0}$, we then define $e^{Gt} := e^{Qt}|_{A^c}$. We have the following result:

**Proposition 5.** *Fix $Q \in \mathcal{Q}$ and let $G$ be its subgenerator. Then $e^{Gt} = \lim_{n \to +\infty}\big(I + {}^t/_n G\big)^n$ for all $t \in \mathbb{R}_{\geq 0}$. Moreover, the family $(e^{Gt})_{t \in \mathbb{R}_{\geq 0}}$ is a semigroup.*

Analogously, we define $\underline{G} := \underline{Q}|_{A^c}$ and $\overline{G} := \overline{Q}|_{A^c}$ to be the *lower-* and *upper subgenerators* corresponding to $\underline{Q}$ and $\overline{Q}$, respectively. We also let $e^{\underline{G}t} := e^{\underline{Q}t}|_{A^c}$ and $e^{\overline{G}t} := e^{\overline{Q}t}|_{A^c}$. Perhaps unsurprisingly, we then have:

**Proposition 6.** *It holds that $e^{\underline{G}t} = \lim_{n \to +\infty}\big(I + {}^t/_n \underline{G}\big)^n$ and $e^{\overline{G}t} = \lim_{n \to +\infty}\big(I + {}^t/_n \overline{G}\big)^n$ for all $t \in \mathbb{R}_{\geq 0}$. Moreover, the families $(e^{\underline{G}t})_{t \in \mathbb{R}_{\geq 0}}$, $(e^{\overline{G}t})_{t \in \mathbb{R}_{\geq 0}}$ are semigroups.*

Our Assumption 2 implies the norm bound:

**Proposition 7.** *For any $t > 0$, it holds that $\|e^{\overline{G}t}\| < 1$.*

It is a straightforward consequence of the use of the supremum norm, together with Equation (7) and the fact that $e^{Qt}$ and $e^{\overline{Q}t}$ are (upper) transition operators, that also $\|e^{Gt}\| \leq \|e^{\overline{G}t}\| < 1$ for all $t \in \mathbb{R}_{>0}$. Hence by the semigroup property we immediately have that $\lim_{t \to +\infty} \|e^{Gt}\| = 0$. This also implies the following well-known result.

**Proposition 8.** *[Taylor and Lay, 1958, Thm IV.1.4] For any $Q \in \mathcal{Q}$ with subgenerator $G$, and all $t > 0$, the inverse operator $(I - e^{Gt})^{-1}$ exists, and $(I - e^{Gt})^{-1} = \sum_{k=0}^{+\infty} e^{Gtk}$.*

This allows us to characterize hitting times for discrete-time homogeneous Markov chains whose transition matrix is given by $e^{Qt}$, as follows.

**Proposition 9.** *Choose any $Q \in \mathcal{Q}$, let $G$ be its subgenerator, and fix any $\Delta > 0$. Let $P \in \mathbb{P}_{\mathbb{N}_0}^{\text{HM}}$ be such that $^P T = e^{Q\Delta}$. Then the expected hitting times $h := \mathbb{E}_P[\tau_{\mathbb{N}_0} \mid X_0]$ satisfy $h|_{A^c} = (I - e^{G\Delta})^{-1}\mathbf{1}$ and $h(x) = 0$ for all $x \in A$.*

*Proof.* By Proposition 1, in $x \in A^c$ we have that

$$h(x) = \mathbb{I}_{A^c}(x) + \mathbb{I}_{A^c}(x)e^{Q\Delta}h(x) = 1 + e^{Q\Delta}h(x).$$

Conversely, it is immediate from the definition that $h(x) = 0$ for all $x \in A$. This implies that $h = (h|_{A^c})\!\uparrow_{\mathcal{X}}$, and hence

$$h|_{A^c} = \mathbf{1} + \big(e^{Q\Delta}(h|_{A^c})\!\uparrow_{\mathcal{X}}\big)|_{A^c} = \mathbf{1} + e^{G\Delta}h|_{A^c}.$$

Re-ordering terms we have $(I - e^{G\Delta})h|_{A^c} = \mathbf{1}$. Now use Proposition 8 and multiply with $(I - e^{G\Delta})^{-1}$. $\qquad\square$

We need the following observation:

**Lemma 1.** *Consider any $Q \in \mathcal{Q}$ with subgenerator $G$, and let $\sigma(G)$ be the set of eigenvalues of $G$. Then $\operatorname{Re}\lambda < 0$ for all $\lambda \in \sigma(G)$.*

This implies that $0 \notin \sigma(G)$, and so we have:

**Corollary 1.** *For any $Q \in \mathcal{Q}$ with subgenerator $G$, the inverse operator $G^{-1}$ exists.*

This allows us to characterize hitting times for continuous-time homogeneous Markov chains:

**Proposition 10.** *Choose any $Q \in \mathcal{Q}$, let $G$ be its subgenerator, and let $P \in \mathbb{P}_{\mathbb{R}_{\geq 0}}^{\text{HM}}$ with $^P Q = Q$. Then the expected hitting times $h := \mathbb{E}_P[\tau_{\mathbb{R}_{\geq 0}} \mid X_0]$ satisfy $h|_{A^c} = -G^{-1}\mathbf{1}$ and $h(x) = 0$ for all $x \in \bar{A}$.*

*Proof.* By Proposition 2, in $x \in A^c$ we have that

$$-1 = -\mathbb{I}_{A^c}(x) = \mathbb{I}_{A^c}(x)Qh(x) = Qh(x).$$

Conversely, it is immediate from the definition that $h(x) = 0$ for all $x \in A$. This implies $h = (h|_{A^c})\!\uparrow_{\mathcal{X}}$, and hence

$$Gh|_{A^c} = \big(Q(h|_{A^c})\!\uparrow_{\mathcal{X}}\big)|_{A^c} = (Qh)|_{A^c} = -\mathbf{1}.$$

Now use Corollary 1 and multiply with $G^{-1}$. $\qquad\square$

### 4.1 QUASICONTRACTIVITY OF SUBSPACE DYNAMICS

We already know from Proposition 7 that $\|e^{\overline{G}t}\| < 1$ for all $t \in \mathbb{R}_{>0}$. Since $e^{\overline{G}0} = I$ (because it is a semigroup), it follows that $\|e^{\overline{G}t}\| \leq 1$ for all $t \in \mathbb{R}_{\geq 0}$. A semigroup that satisfies this property is said to be *contractive*. Moreover, Proposition 7 together with the semigroup property implies that $\lim_{t\to+\infty} \|e^{\overline{G}t}\| = 0$. A semigroup that satisfies this property is said to be *uniformly exponentially stable*, and in such a case the following result holds:

**Proposition 11.** *There are $M \geq 1$ and $\xi > 0$ such that $\|e^{\overline{G}t}\| \leq Me^{-\xi t}$ for all $t \in \mathbb{R}_{\geq 0}$.*

This result means that the norm $\|e^{\overline{G}t}\|$ decays exponentially as $t$ grows. However, for technical reasons we require an exponentially decaying norm bound with $M = 1$; if this holds the semigroup is said to be *quasicontractive*.

It is not clear that obtaining such a bound is possible when $\|e^{\overline{G}t}\|$ is induced by the supremum norm $\|\cdot\|$ on $\mathbb{R}^{A^c}$. However, we can get it by defining a *different* norm $\|\cdot\|_*$ on $\mathbb{R}^{A^c}$. We then obtain the quasicontractivity with respect to the induced operator norm $\|\cdot\|_*$. Because $\mathbb{R}^{A^c}$ is finite-dimensional these norms are equivalent, and such a result suffices for our purposes. This re-norming trick is originally due to Feller [1953], and an analogous construction is commonly used for semigroups of linear operators; see e.g. [Renardy and Rogers, 2006, Thm 12.21].

So, consider the $\xi > 0$ from Proposition 11, and let

$$\|f\|_* := \sup_{t \in \mathbb{R}_{\geq 0}} \left\| e^{\xi t} e^{\overline{G}t} |f| \right\| \quad \text{for all } f \in \mathbb{R}^{A^c}, \quad (8)$$

where $|f|$ denotes the elementwise-absolute value of $f$.

**Proposition 12.** *The map $f \mapsto \|f\|_*$ is a norm on $\mathbb{R}^{A^c}$.*

Moreover, we have the desired result:

**Proposition 13.** *We have $\|e^{\overline{G}t}\|_* \leq e^{-\xi t}$ for all $t \in \mathbb{R}_{\geq 0}$.*

Finally, the same bound holds for precise models:

**Proposition 14.** *For any $Q \in \mathcal{Q}$ with subgenerator $G$ it holds that $\left\|e^{Gt}\right\|_* \leq e^{-\xi t}$ for all $t \in \mathbb{R}_{\geq 0}$.*

## 5 HITTING TIMES AS LIMITS

We now have all the pieces to explain the proof of our main results. The trick will be to establish a connection between hitting times for continuous-time imprecise-Markov chains, and hitting times for *discrete*-time imprecise-Markov chains, for which analogous results were previously established by Krak et al. [2019].

We essentially just look at a discretized continuous-time Markov chain taking steps of some fixed size $\Delta > 0$, derive the expected hitting time for this discrete-time Markov chain, and then take the limit $\Delta \to 0^+$. The main difficulty is in establishing that this converges uniformly for all elements in our sets of processes; this is why we went through the trouble of establishing quasicontractivity in Section 4.1.

To start, for any $Q \in \mathcal{Q}$ and $\Delta > 0$, let $h_\Delta^Q$ be the minimal

non-negative solution to the linear system[8]

$$h_\Delta^Q = \Delta \mathbb{I}_{A^c} + \mathbb{I}_{A^c} e^{Q\Delta} h_\Delta^Q \,, \tag{9}$$

and let $h^Q$ be the minimal non-negative solution to

$$\mathbb{I}_A h^Q = \mathbb{I}_{A^c} + \mathbb{I}_{A^c} Q h^Q \,. \tag{10}$$

Then we know from Propositions 1 and 2 that $1/\Delta h_\Delta^Q$ represents the expected hitting times of a discrete-time homogeneous Markov chain with transition matrix $e^{Q\Delta}$, and that $h^Q$ does the same for a continuous-time homogeneous Markov chain with rate matrix $Q$. We now have the following result:

**Proposition 15.** *There are $\delta > 0$ and $L > 0$ such that* $\left\| h_\Delta^Q - h^Q \right\| < \Delta L \left\| h^Q \right\|$ *for all $\Delta \in (0, \delta)$ and all $Q \in \mathcal{Q}$.*

Since $\left\| h^Q \right\|$ is bounded due to Proposition 10:

**Corollary 2.** *We have* $\lim_{\Delta \to 0^+} h_\Delta^Q = h^Q$ *for all $Q \in \mathcal{Q}$.*

We will now set up the analogous results for imprecise-Markov chains. First, let

$$\underline{h} := \inf_{Q \in \mathcal{Q}} h^Q \quad \text{and} \quad \overline{h} := \sup_{Q \in \mathcal{Q}} h^Q \,. \tag{11}$$

Clearly, it follows from Proposition 2 and the definition of lower- and upper expectations that these quantities represent the lower- and upper expected hitting times for the imprecise-Markov chain $\mathcal{P}_Q^{\mathrm{HM}}$, i.e. it holds that

$$\underline{h} = \underline{\mathbb{E}}_Q^{\mathrm{HM}} \left[ \tau_{\mathbb{R}_{\geq 0}} \,|\, X_0 \right] \quad \text{and} \quad \overline{h} = \overline{\mathbb{E}}_Q^{\mathrm{HM}} \left[ \tau_{\mathbb{R}_{\geq 0}} \,|\, X_0 \right] \,.$$

Now for any $\Delta > 0$, let $\underline{h}_\Delta$ and $\overline{h}_\Delta$ denote the minimal non-negative solutions to the *non-linear* systems

$$\underline{h}_\Delta = \Delta \mathbb{I}_{A^c} + \mathbb{I}_{A^c} e^{\underline{Q}\Delta} \underline{h}_\Delta \tag{12}$$

and

$$\overline{h}_\Delta = \Delta \mathbb{I}_{A^c} + \mathbb{I}_{A^c} e^{\overline{Q}\Delta} \overline{h}_\Delta \,. \tag{13}$$

It was previously shown by Krak et al. [2019] that—up to re-scaling with $1/\Delta$—the quantities $\underline{h}_\Delta$ and $\overline{h}_\Delta$ represent the lower (resp. upper) expected hitting times of, identically, the discrete-time imprecise-Markov chains $\mathcal{P}_{\mathcal{T}_\Delta}^{\mathrm{HM}}$, $\mathcal{P}_{\mathcal{T}_\Delta}^{\mathrm{M}}$, and $\mathcal{P}_{\mathcal{T}_\Delta}^{\mathrm{I}}$ parameterized by the set $\mathcal{T}_\Delta$ of transition matrices that dominate $e^{\underline{Q}\Delta}$. We now set out of prove an analogous result for continuous-time imprecise-Markov chains. We start with the following:

**Proposition 16.** *It holds that* $\lim_{\Delta \to 0^+} \underline{h}_\Delta = \underline{h}$ *and* $\lim_{\Delta \to 0^+} \overline{h}_\Delta = \overline{h}$.

---

[8]Note the re-scaled term $\Delta \mathbb{I}_{A^c}$ on the right-hand side, which distinguishes this from the system in Proposition 1; this is required since the hitting times for discrete-time Markov chains are expressed in the *number* of steps, and to pass to continuous-time we need to measure the size of these steps.

This property allows us to leverage recent results by Erreygers [2021] and Krak [2021] regarding discrete and finite approximations of lower- and upper expectations in continuous-time imprecise-Markov chains, to obtain our first main result:

**Theorem 1.** *It holds that*

$$\underline{h} = \underline{\mathbb{E}}_Q^{\mathrm{HM}} \left[ \tau_{\mathbb{R}_{\geq 0}} \,|\, X_0 \right] = \underline{\mathbb{E}}_Q^{\mathrm{M}} \left[ \tau_{\mathbb{R}_{\geq 0}} \,|\, X_0 \right] = \underline{\mathbb{E}}_Q^{\mathrm{I}} \left[ \tau_{\mathbb{R}_{\geq 0}} \,|\, X_0 \right] \,,$$

*and, moreover, that*

$$\overline{h} = \overline{\mathbb{E}}_Q^{\mathrm{HM}} \left[ \tau_{\mathbb{R}_{\geq 0}} \,|\, X_0 \right] = \overline{\mathbb{E}}_Q^{\mathrm{M}} \left[ \tau_{\mathbb{R}_{\geq 0}} \,|\, X_0 \right] = \overline{\mathbb{E}}_Q^{\mathrm{I}} \left[ \tau_{\mathbb{R}_{\geq 0}} \,|\, X_0 \right] \,.$$

Moreover, it follows relatively straightforwardly from Proposition 16 that the lower- and upper expected hitting times for continuous-time imprecise-Markov chains satisfy an immediate generalization of the system that characterizes the expected hitting times for (precise) continuous-time homogeneous Markov chains. This is our second main result:

**Theorem 2.** *Let $\underline{h}$ and $\overline{h}$ denote the lower- and upper expected hitting times for any one of $\mathcal{P}_Q^{\mathrm{HM}}$, $\mathcal{P}_Q^{\mathrm{M}}$, or $\mathcal{P}_Q^{\mathrm{I}}$. Then $\underline{h}$ is the minimal non-negative solution to the non-linear system $\mathbb{I}_A \underline{h} = \mathbb{I}_{A^c} + \mathbb{I}_{A^c} \underline{Q} \, \underline{h}$, and $\overline{h}$ is the minimal non-negative solution to the non-linear system $\mathbb{I}_A \overline{h} = \mathbb{I}_{A^c} + \mathbb{I}_{A^c} \overline{Q} \, \overline{h}$.*

## 6 SUMMARY & CONCLUSION

We have investigated the problem of characterizing expected hitting times for continuous-time imprecise-Markov chains. We have shown that under two relatively mild assumptions on the system's class structure—*viz.* that the target states are absorbing, and can be reached by any non-target state—the corresponding lower (resp. upper) expected hitting time is the same for all three types of imprecise-Markov chains.

We have also demonstrated that these lower- and upper expected hitting times $\underline{h}$ and $\overline{h}$ satisfy the non-linear systems

$$\mathbb{I}_A \underline{h} = \mathbb{I}_{A^c} + \mathbb{I}_{A^c} \underline{Q} \, \underline{h} \quad \text{and} \quad \mathbb{I}_A \overline{h} = \mathbb{I}_{A^c} + \mathbb{I}_{A^c} \overline{Q} \, \overline{h} \,,$$

in analogy with the precise linear system (5). Indeed, we conclude that the lower- and upper expected hitting times for any of these three types of imprecise-Markov chains, can be fully characterized as the unique *minimal* non-negative solutions to these respective systems.

We aim to strengthen these results in future work to hold with fewer assumptions on the system's class structure.

### Acknowledgements

We would like to sincerely thank Jasper De Bock for many stimulating discussions on the subject of imprecise-Markov chains, and for pointing out a technical error in an earlier draft of this work. We are also grateful for the constructive feedback of three anonymous reviewers.

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
