# OpenReview forum: "Hitting Times for Continuous-Time Imprecise-Markov Chains"
_auai.org/UAI/2022/Conference — UAI 2022 Oral_

### Official Review · Reviewer_gppp · 2022-03-25

**Q2(1) Originality/Novelty:** 4
**Q2(2) Significance/Impact:** 3
**Q2(3) Correctness/Technical Quality:** 3
**Q2(6) Clarity Of Writing:** 3
**Q6 Overall Score:** 6
**Q8 Confidence In Your Score:** 4

**Q1 Summary And Contributions:**

The submission studies the problem of computing expected hitting times in continuous-time imprecise Markov chains. It introduces three different types of imprecise MCs depending on homogeneity and the Markovian property. Due to the nature of imprecise probabilities, only a lower and upper bound for the expected hitting time is given. Under two natural assumptions (that the target is reachable, and hits persist) these bounds are computable and are the same for the three types of imprecise MCs.


**Q2 Assessment Of The Paper:**

More detailed information regarding each of these aspects is given below:

**Q2(5) Reproducibility:**

2: Fair: Key resources (e.g., proofs, code, data) are unavailable but key details (e.g., proof sketches, experimental setup) are sufficiently well-described for an expert to confidently reproduce the main results.

**Q3 Main Strengths:**

A very nice preliminaries section describing the nature of MCs, and a sound (as far as I could verify) technical contribution.

**Q4 Main Weakness:**

Due to space limitations, the authors had to choose details to leave out. They left out the technical part, making the submission less self-contained, in particular avoiding a fundamental notion for the later contribution. That makes some of the claims difficult to understand, and impossible to verify.

**Q5 Detailed Comments To The Authors:**

This submission studies the problem of computing the expected hitting times in continuous-time
imprecise Markov chains. After providing extensive preliminaries, the authors introduce three
different types of imprecise MCs depending on homogeneity and the Markovian property. Due to the
nature of imprecise probabilities, expected hitting times cannot be computed precisely, but only
a lower and upper bound can be obtained. The authors show that under two natural assumptions
(essentially, that the target is reachable, and hits remain observed) these bounds are computable
and, more interestingly, are the same for the three types of imprecise MCs.

The submission is essentially divided in two parts. A very nice, clear, introduction to MCs
accessible to non-experts, followed by an extremely technical (although sound, as far as I could
verify, with a caveat explained below) contribution. Both parts have their advantages, but together
they do not seem to work well. Indeed, the authors explain in full detail some of the basic notions
used (for example, use a couple of lines to introduce the notation of real and natural numbers) at the cost of not having space to explain the difficult parts of the contribution or provide adequate
intuitions. As an important example for the lack of self-containment which damages the submission,
the authors barely cover the use of exponential functions to describe time-homogenous continuous MCs
through matrices, and do not even mention how (or whether) the non-homogeneous case can be
characterised. This deeply damages the submission, because the main contribution based on the three
variants of MCs depends on these characterisations. At that point, the authors simply push it under
the rug mentioning that it can be done. But that is impossible to verify. The authors would have done
better reducing the preliminaries (it pains me to say this, as I really enjoyed reading them), and
investing more space in explaining the core contribution. As it stands, I cannot be fully
enthusiastic about it.

**Q7 Justification For Your Score:**

I am in general positive about the submission, but the lack of the fundamental definition, and the impossibility of verifying the claims to it, makes me drastically reduce my score.

**Q9 Complying With Reviewing Instructions:**

1: Yes.

---

### Official Review · Reviewer_FHcD · 2022-04-03

**Q2(1) Originality/Novelty:** 3
**Q2(2) Significance/Impact:** 3
**Q2(3) Correctness/Technical Quality:** 4
**Q2(6) Clarity Of Writing:** 4
**Q6 Overall Score:** 8
**Q8 Confidence In Your Score:** 4

**Q1 Summary And Contributions:**

This paper provides some results about the hitting times for continuous-time Markov chains when there is imprecision in the probabilistic information about the transition probabilities. The authors consider three possible scenarios: the more intuitive one of sets of time-homogeneous Markov chains, that of sets of (not necessarily time homogeneous) Markov chains and that of sets of imprecise probabilities where epistemic irrelevance holds but not necessarily the Markov condition.

**Q2 Assessment Of The Paper:**

More detailed information regarding each of these aspects is given below:

**Q2(4) Quality Of Experiments (Optional):**

3: Good: The experimental evaluation is adequate, and the results convincingly support the main claims.

**Q2(5) Reproducibility:**

4: Excellent: Key resources (e.g., proofs, code, data) are available and key details (e.g., proof sketches, experimental setup) are comprehensively described for competent researchers to confidently and easily reproduce the main results.

**Q3 Main Strengths:**

Somewhat surprisingly, the authors show that in the three cases (imprecise stochastic processes, imprecise Markov chains and imprecise time-continuous Markov chains) the bounds for the expected hitting times are the same. While the second of these models is somewhat artificial to me, the first and third are quite important, so showing that the behave in the same manner in this particular problem is non-trivial and quite relevant.

As I said before, the paper is well-written, with a lot of attention paid to all details. The technical details are by non-means trivial and the theory is not easy to establish. In spite of these theoretical challenges, I think the authors succeed in conveying the main message of the paper to a non-specialist audience.

**Q4 Main Weakness:**

As I discuss below, perhaps the main assumptions and their implications could be discussed in more details. Also, the intuition behind the proofs is not always given, I guess due to the space limitations.

**Q5 Detailed Comments To The Authors:**

-Since they are instrumental in the proofs, perhaps Assumptions 1 and 2 should be discussed in more detail. I agree that Assumption 1 is quite harmless, but Assumption 2 is quite important, considering the thorn issue that is always conditioning on sets of (lower) probability zero in the field of imprecise probabilities. This would also be important if the work was to be extended towards infinite possibility spaces (although I expect that case to be much more challenging due to a number of other technical issues).

-I am somewhat intrigued about the lack of duality between \underline{h} and \overline{h} mentioned in the conclusions, could you add more details about that?

-The equality between the three types of stochastic processes considered in the paper does not hold in other important scenarios, as discussed in some detail in Krak (2021). Perhaps this can be mentioned, as a discussion of why the models coincide in this context and not in others.

-I found a bit confusing at times the use of TI_A(x) for something where the indicator is not applied on x, as is for instance the case in the proof of Proposition 7 (top of first column, page 11): while this is explained earlier in page 2, and in the end I think everything holds up, perhaps a careful reminder would help the reader.

Minor comments:
-Page 2, last line: I believe x_{n+1} should be x_{t_{n+1}} here.
-Page 4, footnote 2: I guess it should read ‘the product of the functions I_{A^c} and ^P T h_P’ (the first P is missing).
-Page 14, first column: it says Lemma ??, please check (I believe it should be Lemma 7).
-Page 15, display ending in (15): it would be clearer if you write h^* instead of h^*_{\Delta_n_j}, since h^* is what you use at the beginning and at the end.


**Q7 Justification For Your Score:**

I think the authors have done a very serious job that include relevant results in a non-trivial problem. My objections mentioned above are minor, and I believe could easily be addressed with a couple of more pages in the paper.

**Q9 Complying With Reviewing Instructions:**

1: Yes.

---

### Official Review · Reviewer_Erkf · 2022-04-13

**Q2(1) Originality/Novelty:** 3
**Q2(2) Significance/Impact:** 2
**Q2(3) Correctness/Technical Quality:** 3
**Q2(6) Clarity Of Writing:** 4
**Q6 Overall Score:** 6
**Q8 Confidence In Your Score:** 2

**Q1 Summary And Contributions:**

The authors first establish a connection between imprecise Markov chains and a set of stochastic processes. Three types of imprecise Markov chains are introduced by first describing their discrete counterparts and then replacing the transition matrices with rate growth matrices.  Using some results for discrete time imprecise MC from Krak 19, where the step are of fixed size \Delta and then taking the lim \Delta\to 0+, they show that the lower and upper expected hitting time is the
same for all

**Q2 Assessment Of The Paper:**

More detailed information regarding each of these aspects is given below:

**Q2(5) Reproducibility:**

3: Good: Key resources (e.g., proofs, code, data) are available and key details (e.g., proofs, experimental setup) are sufficiently well-described for competent researchers to confidently reproduce the main results.

**Q3 Main Strengths:**

-I think the authors do a great job at explaining the concepts in a well structured way and helping a reader like me (who barely knows the basics).
-I think atleast I learnt some concepts about imprecise Markov chains and connections with set of stochastic process.

**Q4 Main Weakness:**

- The paper does not state its limitations/assumptions explicitly(most papers have this time).
- I have written my comments below
- I think due to finite space, the authors had a dilemma of whether to give the introductory background theory which they felt most readers of this conference would be unfamiliar with, and introducing the main theory and contributions for this paper itself, resulting in a bit of a rough transition for the reader.
- Can the authors motivate Machine learning audience of the value of this work by some practical application, I think this is important since this is a theory paper anyways.


**Q5 Detailed Comments To The Authors:**

Some comments:
- although the writing is mostly great, I found the overloading of notation: T (x,y) variables very confusing( one explanation is that I did not understand the theory well) i.e. when you introduce T as an operator in the form of a matrix: the operator T(x,y) on page 2 left side last paragraph, then you again define it in sec 2.2 page 3 as a Transition matrix T(x,y)-(something I am much more familiar with), are they the same thing, or they can be seen as the same thing from two different subjects etc., anyhow it is not clear to me and I found it very confusing. Adding some explanatory tool for the reader will be helpful


**Q7 Justification For Your Score:**

I have to write, I only knew the basics of discrete Markov chain theory, and so my response is at best a guess.

**Q9 Complying With Reviewing Instructions:**

1: Yes.

---

### Decision · Program_Chairs · 2022-05-15

**Decision:**

Accept (Oral)

**Comment:**

Meta Review: All reviewers agree that the topic and the results are interesting, and that the authors paid a lot of effort into making the paper understandable in the given space. The paper clearly makes a quite strong technical contribution, that is however difficult to summarise in 7 pages.

The ensuing criticism is that the paper has to balance between introducing basic concepts and going into very technical contributions, something that may be hard in a limited space. Authors have managed to achieve this reasonably well, yet the passage between introductory notions and core results remain a bit rough. This is probably the main improvement that could be done if the paper is accepted.